# Multiview Scene Graph

**Juexiao Zhang    Gao Zhu    Sihang Li    Xinhao Liu**
**Haorui Song    Xinran Tang    Chen Feng**✉
New York University
{juexiao.zhang, cfeng}@nyu.edu

## Abstract

A proper scene representation is central to the pursuit of spatial intelligence where agents can robustly reconstruct and efficiently understand 3D scenes. A scene representation is either metric, such as landmark maps in 3D reconstruction, 3D bounding boxes in object detection, or voxel grids in occupancy prediction, or topological, such as pose graphs with loop closures in SLAM or visibility graphs in SfM. In this work, we propose to build *Multiview Scene Graphs* (MSG) from unposed images, representing a scene topologically with interconnected place and object nodes. The task of building MSG is challenging for existing representation learning methods since it needs to jointly address both visual place recognition, object detection, and object association from images with limited fields of view and potentially large viewpoint changes. To evaluate any method tackling this task, we developed an MSG dataset based on a public 3D dataset. We also propose an evaluation metric based on the intersection-over-union score of MSG edges. Moreover, we develop a novel baseline method built on mainstream pretrained vision models, combining visual place recognition and object association into one Transformer decoder architecture. Experiments demonstrate that our method has superior performance compared to existing relevant baselines. All codes and resources are open-source at https://ai4ce.github.io/MSG/.

## 1   Introduction

The ability to understand 3D space and the spatial relationships among 2D observations plays a central role in mobile agents interacting with the physical real world. Humans obtain such spatial intelligence largely from our visual intelligence [26, 45]. When humans are situated in an unseen environment and try to understand the spatial structure from visual observations, we don't perceive and memorize the scene by exact meters and degrees. Instead, we build cognitive maps topologically based on visual observations and commonsense [27, 48]. Given imagery observations, we are able to associate the images taken at the same place by finding overlapping visual clues and identifying the same or different objects from various viewpoints. This ability to establish correspondence from visual perception constitutes the foundation of our spatial memory and cognitive representation of the world. Can we equip AI models with similar spatial intelligence?

Motivated by this question, we propose the task of building a **Multiview Scene Graph (MSG)** to explicitly evaluate a representation learning model's capability of understanding spatial correspondences. Specifically, as illustrated in Figure 1, given a set of unposed RGB images taken from the same scene, this task requires building a *place+object* graph consisting of images and object nodes, where images taken at nearby locations are connected, and the appearances of the same object across different views should be associated together as one object node.

---

✉ Corresponding author. The work was supported in part through NSF grants 2238968 and 2322242, and the NYU IT High Performance Computing resources, services, and staff expertise.

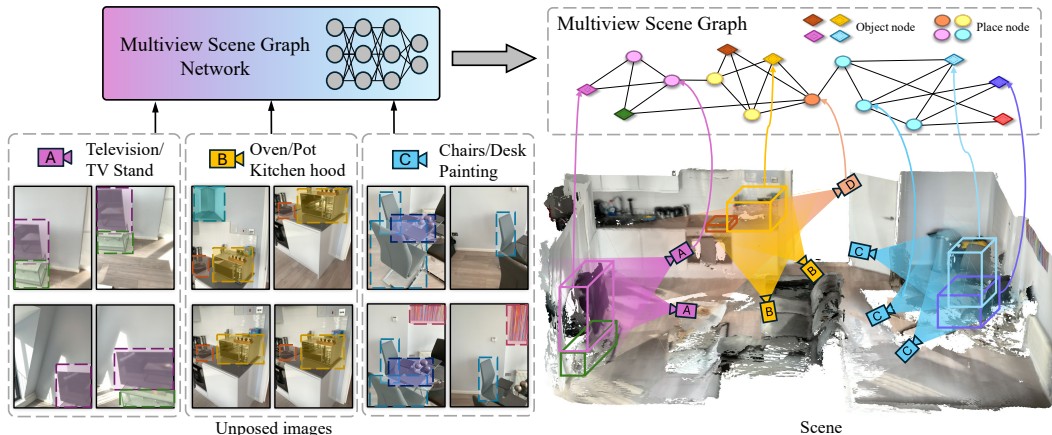

Figure 1: **Multiview Scene Graph (MSG)**. The task of MSG takes unposed RGB images as input and outputs a place+object graph. The graph contains place-place edges and place-object edges. Connected place nodes represent images taken at the same place. The same object recognized from different views is associated and merged as one node and connected to the corresponding place nodes.

We position the proposed Multiview Scene Graph as a general topological scene representation. It bridges the place recognition from robotics literature [3, 4, 36] and the object tracking and semantic correspondence tasks from computer vision literature [20, 23, 64]. Different from previous work in topological mapping that evaluates a method's performance on downstream tasks such as navigation, we propose to directly evaluate the quality of the multiview scene graph, which explicitly demonstrates a model's spatial understanding with correct visual correspondence of both objects and places across multiple views. Moreover, the MSG does not require any metric map, depth, or pose information, making it adaptable to the vast data of everyday images and videos. This also differentiates MSG from the previous work in 2D and 3D scene graphs [5, 32, 35, 69], where they emphasize objects' semantic relationships or require different levels of 3D and metric information.

To facilitate the research of MSG, we curated a dataset from a publicly available 3D scene-level dataset ARKitScenes [8] and designed a set of evaluation metrics based on the intersection-over-union of the graph adjacency matrix. The detailed definition of the MSG generation task and the evaluation metrics are discussed in Section 3.1. Meanwhile, since this task mainly involves solving place recognition and object association, we benchmarked popular baseline methods respectively in place recognition and object tracking, as well as some mainstream pretrained vision foundation models. We also designed a new Transformer-based architecture as our method, Attention Association MSG, dubbed *AoMSG*, which learns place and object embeddings jointly in a single Transformer decoder and builds the MSG based on the distances in the learned embedding space. Our experiments demonstrate the superiority of our new model compared with the baselines by a great margin, yet still reveal strong needs for future advances in research for spatial intelligence.

In summary, our contributions are two-fold:

- We propose the Multiview Scene Graph (MSG) generation as a new task for evaluating spatial intelligence. We curated a dataset from a publicly available 3D scene dataset and designed evaluation metrics to facilitate the task.
- We design a novel Transformer decoder architecture for the MSG task. It jointly learns embeddings for places and objects and determines the graph according to the embedding distance. Experiments demonstrate the effectiveness of the model over existing baselines.

## 2 Related work

**Scene Graph**    Scene graphs [35, 69] are originally proposed to represent the spatial and semantic relationships between objects in an image. The generated scene graph can be used for image captioning [47] and image retrieval [35]. Although they provide a structured spatial representation, it remains at the 2D image level. 3D scene graphs [5, 32, 33, 66, 71] extend this concept into 3D, representing a scene as a topological graph with objects, rooms, and camera positions as their nodes.

These graphs are typically built by abstracting from 3D meshes, point clouds, or directly from RGB-D images. [67] proposes incrementally building 3D scene graphs from RGB sequences, describing semantic relationships between objects. As a new type of scene graph, MSG is built from unposed images without sequential order, emphasizing the understanding of relationships between objects and places via multiview visual correspondences. MSG complements existing scene graphs, as their object-object relationship edges can be a seamless add-on to extend MSG with more semantic information. Therefore, we believe MSG provides a meaningful contribution to the scene graph community by enhancing its representational depth and flexibility.

**Scene Mapping**    Simultaneous localization and mapping (SLAM) [17, 46, 57, 59] is a classic way of creating maps of an environment from observations. The metric maps built from SLAM are subsequently utilized as the spatial representation for the robots to perform tasks such as navigation. In contrast to metric maps, topological mapping  [56] is inspired by landmark-based memory in animals, and follows a more natural and human-like understanding of the environment to better support navigation tasks. The quality of the topological maps is evaluated mostly through navigation tasks [11, 15]. Another line of scene mapping work harnesses object or semantic information to build more robust maps [25, 54, 68], with TSGM [37] being the most relevant to our work. Differently, our proposed MSG serves as a general-purpose scene representation and can be directly evaluated using our proposed metrics. The quality of MSG that a model can build explicitly evaluates its capability of understanding spatial correspondences.

**Visual Place Recognition**    Visual Place Recognition (VPR) is often formed as an image retrieval problem. This involves extracting image features and retrieving the closest neighbor from an image database. Traditional approaches rely on handcrafted features [9, 41]. NetVLAD and its variants[4, 16, 44] use deep-learned image features to improve recall performance and robustness. The emergence of self-supervised foundation models, such as DINOv2 [49], enables universal image representations, offering significant progress [34, 36] across many VPR tasks. However, VPR is framed as an image retrieval problem, whose output—the image features—does not directly equal a graph. Although a graph can be built by proximity search in the VPR feature space, the widely used recall metric in VPR does not directly reflect how good the graph is, i.e. how many pairs of connected images in this graph are truly at the same place. Instead, our proposed task and evaluation metric focus only on the graph generated from the model. The metric straightfordly reflects the quality of the scene representation.

**Object Association**    Traditionally, object association is approached by matching keypoint features across image pairs [41, 55]. Recently, CSR [24] learns feature encodings of object detections and measures the cosine similarity between the learned features to determine object matching. ROM [23] on the other hand follows SuperGlue [55] and uses attentional GNN and Sinkhorn distance [58] for relational object matching. Our method draws inspiration from this previous work but adopts a Transformer decoder architecture and learns object instance embeddings jointly in a unified model with place recognition. Literature in multi-object tracking [50, 64, 65, 72] and video object segmentation [19, 20, 31, 62] also handles object association. They mostly leverage temporal dependencies or memories such as by propagating detection bounding boxes or segments through time. Therefore, these models may lack a sense of space and suffer when objects reappear from a very different viewpoint or after a longer period. Interestingly, a recent study Probe3d [22] reveals that even though the pretrained vision foundation models have undergone tremendous progress in the recent years [14, 21, 30, 38, 49], they still struggle with associating spatial correspondences of objects from large viewpoint change. Our method learns scene representation with spatial correspondence, where multiple views of the same places or the same objects are close in the embedding space.

## 3   Multiview scene graph

### 3.1   Problem definition

**Multiview Scene Graph**    Given a set of unposed images of a scene $X = \{x_i\}_{i=0,...,T}$, we represent a Multiview Scene Graph as a **place+object graph**:

$$G = \{P, O, E^{PP}, E^{PO}\}, \tag{1}$$

where $P$ and $O$ respectively refer to the sets of *place* and *object* nodes. The set of object nodes $O$ contains all the objects detected from $X$. The same object detected from different images across different viewpoints should always be considered as one object node. For the definition of places, we follow the definition in the VPR literature and set $P = X$. This means each image corresponds to a node for a place, and if two images are taken within only a small translation and rotation distance, they are considered as taken in the same place and are connected with an edge in $E^{PP}$. Consequently, the $E^{PP}$ is the set of *place-place edges* which refers to the edges that connect the images regarded as in the same place, and the $E^{PO}$ represents the set of *place-object edges*, referring to the edges that connect the places and the objects that appear in those places. Therefore, an object can be seen in multiple images and thus connected to more than just one place node. These images can be close by or from a distance. Naturally, a place node can connect to more than one object node, since an image can contain multiple objects' appearances.

**MSG generation task**  As illustrated in Figure 1, the MSG generation task requires building an estimated place+object graph $\hat{G}$ from the unposed RGB image set. The graph is further represented as a place+object adjacency matrix $\hat{A}$ of size $(|P| + |\hat{O}|) \times (|P| + |\hat{O}|)$, while the groundtruth $G$ is represented by $A$ of size $(|P| + |O|) \times (|P| + |O|)$. Note that the object set $\hat{O}$ may differ from $O$. The quality of $\hat{G}$ is evaluated by measuring $\hat{A}$ against the groundtruth $A$. According to our definition, the adjacency matrix can be further decomposed into the following block matrix:

$$A = \left[ \begin{array}{cc} A^{PP} & A^{PO} \\ A^{OP} & A^{OO} \end{array} \right], \tag{2}$$

where $A^{PP} = A_{1 \leq i \leq |P|, 1 \leq j \leq |P|}$ and $A^{PO} = A_{1 \leq i \leq |P|, |P|+1 \leq j \leq |P|+|O|}$ . The same decomposition applies to $\hat{A}$. Since the MSG contains only the place-place edges and the place-object edges, $A^{OO}$ is left blank. Meanwhile, $A^{PO}$ is symmetric to $A^{OP}$. So our evaluation will focus on $A^{PP}$ and $A^{PO}$.

## 3.2  Evaluation metric

Given that the two adjacency matrices $A$ and $\hat{A}$ are binary, we evaluate their intersection over union (IoU) to measure how much the two graphs align. As aforementioned, an adjacency matrix $A$ essentially consists of two parts: the place-place part $A^{PP}$ and the place-object part $A^{PO}$. So we evaluate them respectively as PP IoU and PO IoU and combine them to get the whole graph IoU. We provide a precise mathematical definition of the IoU calculation for any two binary adjacency matrices in Appendix B.1 and we denote this function by $\text{IoU}(\cdot, \cdot)$ in the following for simplicity.

**PP IoU**  For the PP IoU, the calculation is relatively straightforward since the number of images is deterministic and the one-to-one correspondence between the groundtruth $A^{PP}$ and the prediction $\hat{A}^{PP}$ is fixed. As a result, the PP IoU is simply:

$$\text{PP IoU} = \text{IoU}(A^{PP}, \hat{A}^{PP}). \tag{3}$$

Additionally, we also report the Recall@1 score alongside PP IoU since it is the standard evaluation metric for visual place recognition.

**PO IoU**  However, it is less straightforward for the PO IoU. The number of objects in the predicted set $\hat{O}$ may differ from $O$, and their correspondence cannot be determined directly from the adjacency matrix. For a fair evaluation, we need to align $\hat{O}$ with $O$ as much as possible. In other words, before computing IoU, we need to find the best matching object for each groundtruth object. This truth-to-result matching is also an important issue in multi-object tracking [53]. To do so, we also record the object bounding boxes in each image and calculate the generalized IoU score (GIoU) of the bounding boxes following [52]. Then we compute a one-to-one matching between $O$ and $\hat{O}$ based on the accumulated GIoU score across all the images. Details of the score computation are included in the Appendix B.2. According to the matching, we can reorder $\hat{O}$ to best align with the objects in $O$. This can be mathematically expressed as a permutation matrix $S \in \mathbb{R}^{|\hat{O}| \times |\hat{O}|}$ to permute the columns of $\hat{A}$. Formally, the PO IoU is expressed as the following:

$$\text{PO IoU} = \text{IoU}(A^{PO}, \hat{A}^{P\hat{O}} S). \tag{4}$$

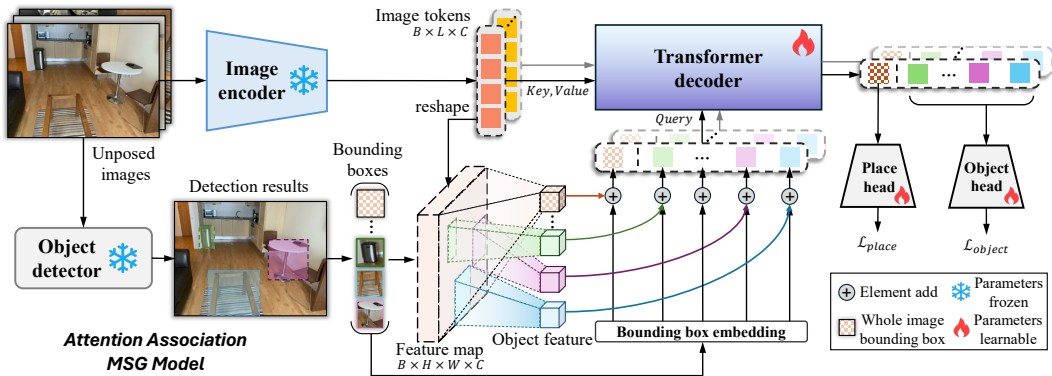

Figure 2: **The AoMSG model.** Places and objects queries are obtained by cropping the image feature map using corresponding bounding boxes. The queries are then fed into the Transformer decoder to obtain the final places and objects embeddings. Bounding boxes are in different colors for clarity. The parameters in the Transformer decoder and the linear projector heads are trained with supervised contrastive learning. Image encoder and object detector are pretrained and frozen.

## 4 Our Baseline: Attention Association MSG Generation

When developing a new model for the MSG generation task, we adhere to two core principles: Firstly, the model should capitalize on the strengths of pretrained vision models. These pretrained models offer a robust initialization for subsequent vision tasks, as their output features encapsulate rich semantic information, forming a solid foundation for tasks like ours. Secondly, both place recognition and object association fundamentally address the problem of visual correspondence and can mutually reinforce each other through contextual information. Thus, our model is designed to integrate both tasks within a unified framework. With these guiding principles, we propose the Attention Association MSG (AoMSG) model, depicted in Figure 2.

**Place and object encodings**  Given a batch of unposed RGB images as input, the AoMSG model first employs pretrained vision encoders and detectors to derive image tokens and object detection bounding boxes from each image. We utilize the Vision Transformer-based pretrained model DINOv2 [49] as our encoder, though our design is adaptable to any Transformer-based or CNN-based encoder that produces a sequence of tokens or a feature map. In the case of the DINOv2 encoder, we reshape the output token sequences into a feature map, which is then aligned to the object bounding boxes, aggregating an encoding feature for each detected object. To integrate place recognition and object association within a unified framework, we obtain the place encoding feature by treating it as a large object with a bounding box that encompasses the entire image, aggregating features as if a detected object. The obtained place feature is then positioned alongside the object features, serving as queries for the Transformer decoder, as detailed in the subsequent sections.

**AoMSG decoder**  We follow a DETR-like structure [13] to design our AoMSG decoder. Specifically, the derived place feature and object features are stacked as a sequence of queries for the Transformer decoder, while the preceding image tokens are used as keys and values. As shown in Figure 2, we enhance the queries by incorporating positional encodings by normalizing and embedding the bounding box coordinates. For instance, for the place feature, the equivalent bounding box is the entire image as aforementioned, resulting in the normalized coordinates of [0, 0, 1, 1]. These coordinates are projected to match the dimensionality of the encoding and added elementwise to the place query. The outputs of the AoMSG Transformer decoder are the place and object embeddings that have aggregated context information from the image tokens. Then two linear projector heads are applied to each object and place embeddings respectively to obtain the final object and place embeddings, projecting them into the representation space for the task.

**Losses and predictions**  For training, we compute supervised contrastive learning [51] respectively on the place and object embeddings from the same training batch in a multitasking fashion. For the object loss, we simply use binary cross-entropy with higher positive weights. For the place loss,

Table 1: **Main results.** Our method uses DINOv2[49] as the backbone. GDino stands for the detector GroundingDINO[39]. AoMSG-2 and AoMSG-4 represent AoMSG models with 2 and 4 layers of Transformer decoder respectively. The best results are underlined. * indicates a trivial result since its input is given in temporal order, and consecutive frames are trivially recalled.

| Method | | Metric | | | |
|---|---|---|---|---|---|
| Place | Object | Recall@1 | PP IoU | PO IoU | |
| | | | | w/ GT detection | w/ GDino [39] |
| AnyLoc [36] | - | 97.1 | 34.2 | - | - |
| NetVlad [4] | - | 96.6 | 35.5 | - | - |
| Mickey [7] | - | 100* | 33.1 | - | - |
| SALAD [34] | - | 97.1 | 35.6 | - | - |
| - | UniTrack [64] | - | - | 17.4 | 13.0 |
| - | DEVA [20] | - | - | 16.2 | 16.6 |
| SepMSG - Direct | | 96.0 | 31.4 | 50.4 | 24.5 |
| SepMSG - Linear | | 96.9 | 34.9 | 59.3 | 24.6 |
| SepMSG - MLP | | 94.3 | 29.2 | 56.9 | 23.4 |
| AoMSG-2 | | 97.2 | 40.7 | 69.1 | 28.1 |
| AoMSG-4 | | 98.3 | 42.2 | 74.2 | 28.1 |

Table 2: **Comparison of different projector dimensions** in AoMSG and SepMSG models. Both are using DINOv2-base[49] as the backbone. Results are evaluated at 30 epochs.

| Projector dimension | AoMSG-4 | | | SepMSG-Linear | | |
|---|---|---|---|---|---|---|
| | Recall@1 | PP IoU | PO IoU | Recall@1 | PP IoU | PO IoU |
| 512 | 97.7 | 41.3 | 72.9 | 93.2 | 20.3 | 59.2 |
| 1024 | 98.3 | 42.2 | 74.2 | 96.9 | 34.9 | 59.3 |
| 2048 | 97.9 | 41.8 | 72.4 | 96.5 | 35.0 | 58.9 |

the mean square error is minimized for their cosine distances, which gives better empirical results. During inference, we simply compute the cosine similarity among the place embeddings and apply a threshold to obtain the place-place predictions in $\hat{A}$. For the objects, we track their appearances and maintain a memory bank of the existing objects for each scene, updating their embeddings or registering new objects based on cosine similarity and thresholding. The results are consequently converted to the place-object part in $\hat{A}$. Notably, there could be many possible choices to compute the contrastive losses and determine the predictions, we keep our choices simple as we empirically find the standard losses and the simple cosine thresholding can already produce decent results while keeping the embedding spaces straightforwardly meaningful. We discuss the results in detail in Section 5.

## 5 Experiment

### 5.1 Data

The MSG models can be trained with any dataset that provides camera poses and object instance labels. We utilized the publicly available 3D indoor scene dataset ARKitScenes [8] to construct our dataset. ARKitScenes contains point clouds and 3D object bounding boxes of the scenes, as well as the calibrated camera poses obtained from an iPad Pro. We transform the point clouds in the 3D bounding boxes with respect to the camera poses to obtain the 2D bounding boxes in each frame. The resolution of each frame is $192 \times 256$. 4492 scenes are used for training and 200 scenes are used for testing. Note that none of the two scenes share the same objects. We leverage the camera poses to obtain the place annotations. Translation threshold and rotation threshold are set to 1 meter and 1 radian respectively, images taken within both thresholds are considered as capturing the same place.

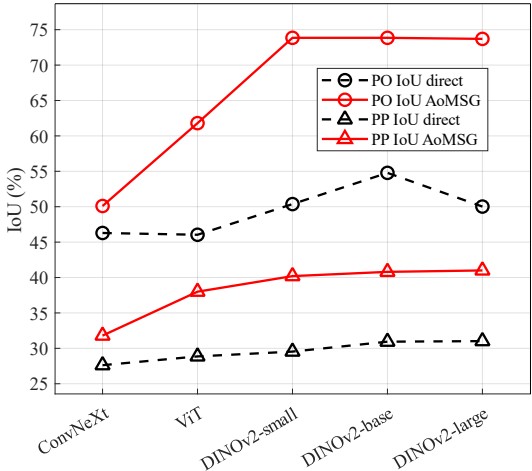

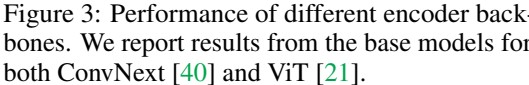

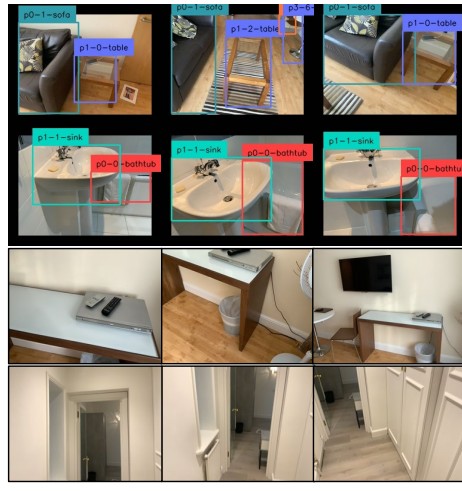

Figure 3: Performance of different encoder back-bones. We report results from the base models for both ConvNext [40] and ViT [21].

Figure 4: Visualization of the same objects and the same places. Objects are annotated with their predicted IDs.

## 5.2 Baselines

**VPR** We adopt protocols in the previous VPR benchmark literature as outlined in [10, 36]. In our off-the-shelf baselines, we evaluated VPR using DINOv2 [49] either as the global descriptor or followed by a VLAD dictionary generated from a large-scale indoor dataset following [36], or as a feature extraction backbone [34]. For the trained baseline, we conduct our experiments mainly with ResNet-50 [28] + NetVLAD used in [10]. Additionally, we also test a recent pose estimation baseline [7] and use the poses to estimate the places according to the same thresholds as in the dataset.

**Object association** We adopt two popular baselines for object association, Unitrack [64] from multi-object tracking, and DEVA [20] from video object segmentation. The image sets are processed in temporal order just like tracking. Unitrack can take any detection backbones and associate object bounding boxes by comparing their features with an online updating memory bank. For a fair comparison, we extend its memory buffer length to cover the whole set of images for every scene. DEVA leverages the Segment Anything model [38] to segment and track any object throughout a video without additional training. Their tracking results can be easily converted for evaluating object association based on the tracker IDs.

**SepMSG** We also evaluate the pretrained vision models by first separately encoding images and object detections to features and directly evaluating MSG based on those features. This baseline is referred to as *SepMSG-Direct*, where *Sep* means *separately* handling places and objects. Then as a common way of evaluating pretrained models [29, 30], we conduct probing [2] by further training a linear or MLP classifier on those frozen features. These baselines are referred to as *SepMSG-Linear* and *SepMSG-MLP*. The SepMSG baselines serve as ablation to validate our model against simply using features learned from the pretrained backbones.

## 5.3 Experimental setups

For AoMSG, we experimented with different choices of backbones, sizes of the Transformer decoder, and dimensions of the final linear projector heads. Their results are discussed in Section 5.4. All the models are trained on a single H100 or GTX 3090 graphics card for 30 epochs or until convergence. We provide detailed hyperparameters in the appendix. During training, we randomly shuffle the scenes and mix data from multiple scenes in a single batch so that the model sees diversified negative samples at every epoch. Additionally, we monitor the total coding rate as in [60] to avoid the embeddings from collapsing.

To keep the evaluation focused on the quality of the graph rather than the quality of object detection, we choose not to train the detector together with the MSG objectives. Instead, we use the groundtruth

detection bounding boxes and a popular open-vocabulary object detector GroundingDINO [39]. Results on both configurations are listed in Table 1 and discussed in the following.

## 5.4 Results

**Main results**    Table 1 shows comparison of our results and baselines. We find that for the place Recall@1 and PP IoU, the baselines have competitive performance. While the results from the SepMSG baselines are comparable, AoMSG outperforms them all and produces the best results in both metrics. We also notice that all the models produce high Recall@1, but their PP IoU scores are varied and less than 50. This suggests that having high recall is not enough to guarantee a good graph. For PO IoU, AoMSG models outperform all the baselines by big margins. Both Unitrack and DEVA perform poorly as they struggle when objects reappear after large viewpoint changes or long periods of time. We note that all the MSG methods produce relatively worse results when using GroundingDINO as the detector rather than the ground truth detection. This indicates the performance gap caused by inaccurate object detection. Nevertheless, their performances are still consistent and AoMSG still performs the best. This suggests a better detector will likely give better results for the MSG task. To conclude, AoMSG gives the best performance for all the metrics.

**Projector dimensions**    As listed in Table 2, we compared the impact of different projector dimensions as it is reportedly important to performance in the literature of self-supervised representation learning [6, 12, 18]. We find the empirical results are comparable in our experiments.

**Choices of backbones**    Figure 3 shows the performance of difference choices of pretrained backbones. We experimented with state-of-the-art CNN-based model ConvNext [40], Vision Transformer (ViT) [21] and DINOv2 [49]. We find DINOv2 performs the best, consistent with the observations made in [22]. We use DINOv2 as our default encoder. Interestingly, performance saturates with the size of DINOv2. We suspect it will still increase if we could further scale the size of the data.

**Qualitative**    In Figure 5 we visualized the learned object embeddings on 6 scenes by AoMSG, the SepMSG-Linear baseline, and SepMSG-Direct that directly uses the output features from the pretrained DINOv2 encoder for the task. The visualization aims to qualitatively assess the learned object embeddings as to how separated different objects are in the embedding space. We can see the pretrained embeddings already provide some decent separations. SepMSG-Linear only tunes a linear probing classifier on top so the separation is slightly improved. For example, see the first and second scenes to the left. Compared with them, AoMSG gives the most significant separations, with appearances of the same objects pushed closely and different objects pulled far away. Additionally, Figure 4 visualizes results on some places and objects, and we provide more in Appendix D.

## 6    Discussion

### 6.1    Application

Given the recent advances in novel view synthesis, 3D reconstruction, and metric mappings, one might wonder whether the proposed MSG is still useful. Here we provide some justifications and a showcase application. Echoing literature in 3D scene graphs [32, 66], we believe the MSG can be a versatile mental model for embodied AI agents and robots. At a global level, it keeps a lightweight topological memory of the scene from purely 2D RGB inputs, which serves as a basis for robot navigation [15, 37]. At a finer level, it can seamlessly couple MSG with the 3D reconstruction methods, to estimate depth and poses and build local reconstructions. Therefore, a robot can traverse the environment, localize itself referring to the MSG, and build a local reconstruction when needed for tasks that require metric information such as the manipulation tasks.

As a showcase application, we provide two local 3D reconstruction cases illustrated in Figure 6 using the most recent off-the-shelf 3D reconstruction model Dust3r [63]. Directly applying Dust3r to a dense image set greatly consumes GPU memory, which may be infeasible for mobile robots. Whereas a random subsample does not guarantee the reconstruction quality. Instead, with MSG, we can provide the Dust3r with locally interconnected subgraphs for fast and reliable local reconstruction. The subgraphs and local reconstructions can be object-centric thanks to the *place+object* nature of

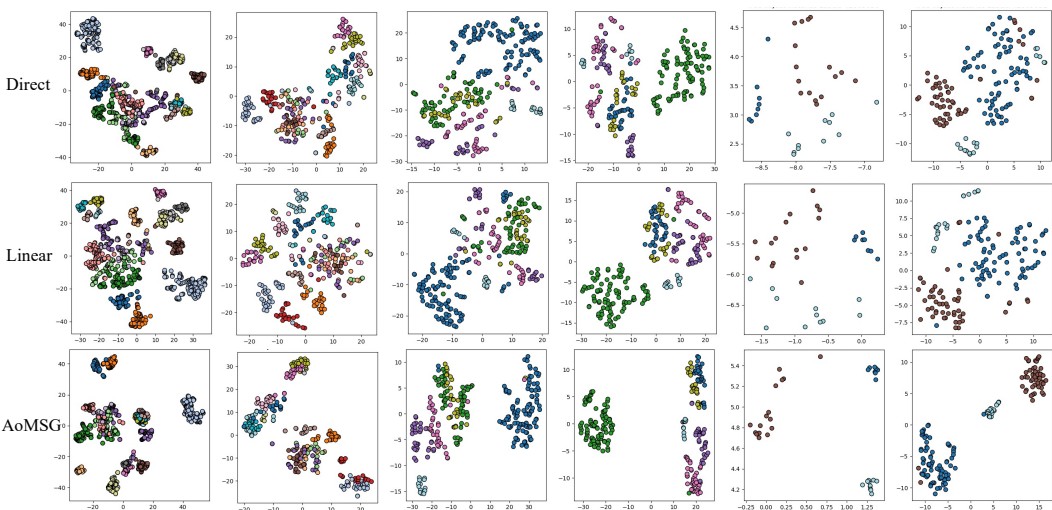

Figure 5: Object embedding visualization using t-SNE [61]. SepMSG-Direct, SepMSG-Linear, and AoMSG-2 are shown in each row respectively. Results from the same scene are aligned vertically. Colors indicate different objects. Each point is an appearance of an object. It is best viewed in color.

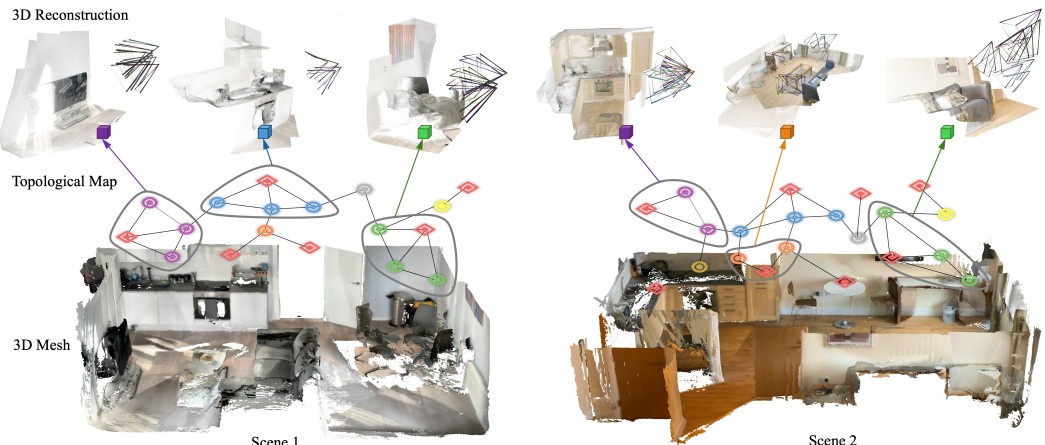

Figure 6: Local 3D reconstruction from 2D MSG using off-the-shelf model Dust3r [63]. The 3D meshes of two scenes are shown side by side, with 3 subgraphs circled in gray and reconstructed on the top of each scene.

MSG. Moreover, the local reconstructions are topologically connected by MSG. This suggests MSG can provide a flexible scene representation balancing 2D and 3D, abstractions and details.

## 6.2 Limitation

The current work still has many limitations. Firstly, we only conducted experiments in one dataset. Although the dataset contains around 5k scenes, which is sufficient to obtain convincing results, it would still be great to see if training on more diversified data collections can produce better models and stronger generalization as observed in [63], especially for larger models. We leave this to future work. Secondly, scenes in the current dataset contain only static objects, extending to dynamic objects is a direction worth exploring.

Additionally, given the scope of the work is to propose MSG as a new vision task promoting spatial intelligence, we focus on explicitly evaluating the quality of the graph. Therefore, we did not

investigate the object detection quality, nor did we deploy the MSG to downstream tasks such as navigation. Note that detection quality does affect the MSG performance though we find it to be consistent across different detection modes, i.e. the groundtruth and the GroundingDINO. Training detectors together with the MSG model and applying MSG to downstream tasks will be our next step to make the work a more complete system.

## 7 Conclusion

This work proposes building the Multiview Scene Graph (MSG) as a new vision task for evaluating spatial intelligence. The task gives unposed RGB images as input and requires a model to build a place+object graph that connects images taken at the same place and associates the object recognitions from different viewpoints, forming a topological scene representation. To evaluate the MSG generation task, we designed evaluation metrics, curated a dataset, and proposed a new model that jointly learns place and object embeddings and builds the graph based on embedding distances. The model outperforms existing baselines that handle place recognition and object association separately. Lastly, we discussed the possible applications of MSG and the current limitations. We hope this work can stimulate future research on advancing spatial intelligence and scene representations.

**Acknowledgement.** The authors thank Yiming Li and Shengbang Tong for their valuable discussions and suggestions.

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

# A  Hyperparameter setting

| Hyperparameter | Value/Range |
|---|---|
| Original image size | $192 \times 256$ |
| Input image size | $224 \times 224$ |
| Batch size | 384 |
| Scenes per training batch | 6 |
| Images per scene per batch | 64 |
| Learning rate | 2e-5 |
| Epochs | 30 |
| Optimizer | AdamW |
| Scheduler | None |
| Weight decay | 0.01 |
| AoMSG layers | 2, 4 |
| AoMSG patch size | 14 |
| AoMSG hidden dim | 384 |
| Projector head dim | 512, 1024, 2048 |
| $L_{place}$ Place Loss Function | MSE on cosine |
| $L_{object}$ Object Loss Function | BCE w/ positive weight=10 |
| Loss ratio $L_{place} : L_{object}$ | 1: 1 |
| Place Similarity Threshold | 0.3 |
| Object Similarity Threshold | 0.2 |

Table 3: Hyperparameters used in the AoMSG main experiments.

# B  Details of evaluation metrics

## B.1  IoU between two adjacency matrices

Given two binary adjacency matrices $A \in \{0,1\}^{m_A \times n_A}$ and $B \in \{0,1\}^{m_B \times n_B}$. Suppose the vertices in their corresponding graphs have been compared and best-matched, we can directly compute the IoU as the following:

$$m^* = \min(m_A, m_B) \tag{5}$$

$$n^* = \min(n_A, n_B) \tag{6}$$

$$w_A = \sum_{m^* < i \leq m_A, n^* < j \leq n_A} A_{ij} \tag{7}$$

$$w_B = \sum_{m^* < i \leq m_B, n^* < j \leq n_B} B_{ij} \tag{8}$$

$$\text{IoU}(A, B) = \frac{\sum_{1 \leq i \leq m^*, 1 \leq j \leq n^*} A_{ij} \wedge B_{ij}}{\sum_{1 \leq i \leq m^*, 1 \leq j \leq n^*} A_{ij} \vee B_{ij} + w_A + w_B}. \tag{9}$$

The inclusion of $w_A$ and $w_B$ in the denominator implies that the IoU value decreases when the two graphs contain additional but not isolated vertices.

From the graph perspective, this IoU evaluates binary edge prediction on augmented graphs. Specifically, given a groundtruth graph and a predicted graph, after matching their vertices as discussed in Sec 3.2, we can augment both graphs with unmatched vertices from the other. We name eq(9) as IoU because under this construction, the defined IoU aligns with the following definition in binary classification based on the standard True Positives (TP), False Negatives (FN), and False Positives (FP):

$$\text{IoU} = \frac{\text{TP}}{\text{TP} + \text{FP} + \text{FN}}. \tag{10}$$

Here, a "positive" prediction indicates the presence of an edge.

## B.2 Object truth-to-result matching

Here we elaborate on the computation of the object matching score mentioned in Sec. 3.2.

Given any groundtruth object $\forall \gamma \in O$ and any predicted object $\forall \tau \in \hat{O}$, we record and compare their detections across all the $T$ frames. Denote $D(\gamma, t)$ and $D(\tau, t)$ as the groundtruth and predicted object detections in frame $t$ for $\gamma$ and $\tau$ respectively. Use the indicator functions $\mathbb{I}(\cdot, t)$ to signal the existence of an object in frame $t$: if $\tau$ exists in $t$, then $\mathbb{I}(\tau, t) = 1$, otherwise $\mathbb{I}(\tau, t) = 0$.

Therefore, the accumulated GIoU of $\gamma$ and $\tau$ is:

$$c_{\gamma,\tau} = \sum_{t \in T} \mathbb{I}(\gamma, t) * \mathbb{I}(\tau, t) * \text{GIoU}\left(D(\gamma, t), D(\tau, t)\right), \tag{11}$$

which is the sum of the generalized bounding box IoU [52] across all the frames that both objects are present. To obtain the final matching score between $\gamma$ and $\tau$, $c_{\gamma,\tau}$ is further normalized by the sum of the following four terms:

- The number of the matched frames, where both $\gamma$ and $\tau$ exist and their GIoU is positive.
- The number of the unmatched frames, where both $\gamma$ and $\tau$ exist but their GIoU is zero.
- The number of the "false positive" frames, where only $\tau$ exists and $\gamma$ doesn't.
- The number of the "false negative" frames, where only $\gamma$ exists and $\tau$ doesn't.

The sum of these four terms is in fact equivalent to computing the union of the appearances of $\gamma$ and $\tau$ across all the $T$ frames:

$$u_{\gamma,\tau} = \sum_{t \in T} \mathbb{I}(\gamma, t) + \sum_{t \in T} \mathbb{I}(\tau, t) - \sum_{t \in T} \mathbb{I}(\gamma, t) * \mathbb{I}(\tau, t). \tag{12}$$

Consequently, the matching score between $\gamma$ and $\tau$ is computed as:

$$m_{\gamma,\tau} = \frac{c_{\gamma,\tau}}{u_{\gamma,\tau}}. \tag{13}$$

In practice, we take $1 - m_{\gamma,\tau}$ as the cost used in solving the one-to-one assignment problem via Hungarian Matching.

## C  Additional Analysis

### C.1  Learned relative pose distributions

We set thresholds as the dataset hyperparameters which is a conventional setup in visual place recognition (VPR) tasks and datasets [42, 70]. Since the MSG task involves place recognition, we choose to adopt this convention. VPR tasks require a model to classify whether or not two images are taken from the same place. The concept of "place" is a discretization of the space that is continuous by nature, necessitating the use of thresholds in the VPR setup.

To give a closer look at the effect the threshold has on the model, in Figure 7 we report the relative pose distributions (orientation and translation) for the connected and non-connect nodes based on our model's prediction. The figures show that instead of collapsing to only represent the fixed thresholds, the pose distributions have a clear yet smooth separation across the spatial thresholds.

### C.2  Failure cases

We visualize some failure cases for place recognition in Figure 8. We observe that most failure cases can be attributed to either having very similar visual features with relatively large pose differences (false positives), such as observing a room from two opposite sides, or having few similarities in visual features with relatively smaller pose differences (false negatives). We note that the recall metric, conventional in VPR, is straightforward and effective for image retrieval against a database. However, it falls short in reflecting challenging false positives and negatives, especially when constructing a topological graph like the MSG where the number of positives varies. This highlights the usefulness of our proposed IoU metric, which consistently evaluates the quality of the graph.

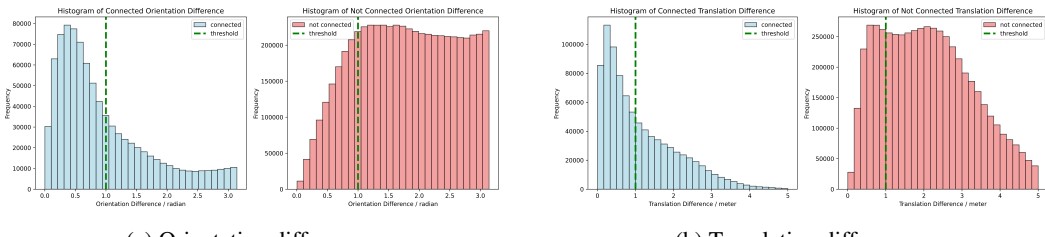

(a) Orientation difference.       (b) Translation difference.

Figure 7: **Relative pose distribution** in histograms on the test set. Blue is for the connected and red is for the not connected. The green dashed lines are the spatial thresholds.

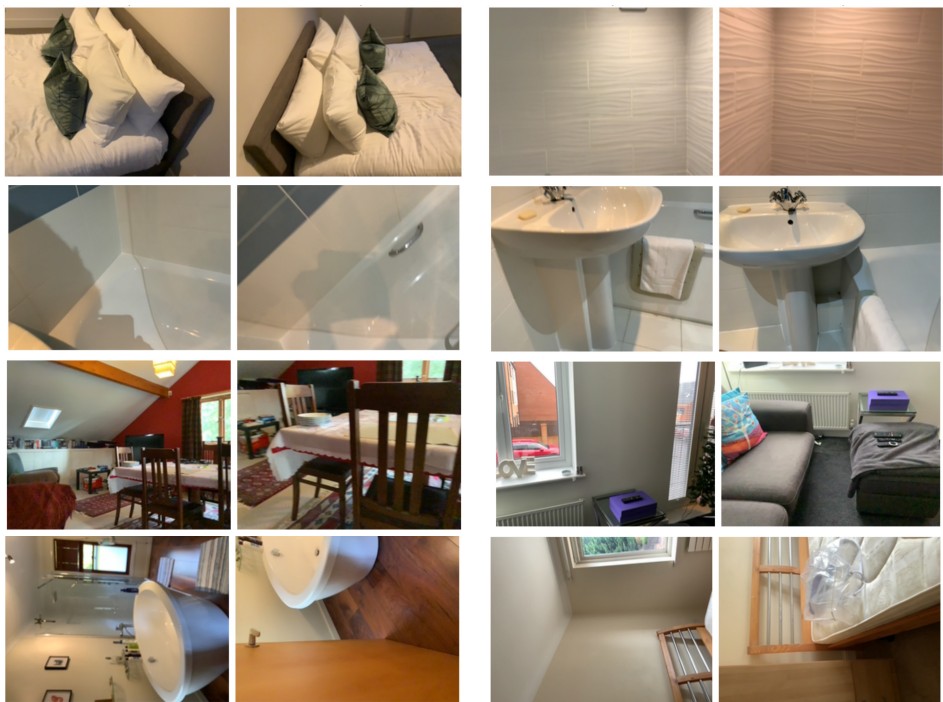

Figure 8: **Failure cases for place recognition.** The top 2 rows are false positive and the bottom 2 rows are false negative. Listed in pairs.

### C.3 Approaching MSG Generation with Multimodal Large Language Model

Multimodal Large Language Models (MLLMs) have exhibited strong emergent abilities for many tasks, and we are intrigued to try them out for the MSG generation task. However, querying MLLMs with every image pair in an image set is a huge amount of work and cost, and sending all images together poses a challenge to the context length limit while also hurting performance. Therefore, we conducted a case study with one scene as a pilot study.

Specifically, we sampled a scene with a relatively small number of images and further subsampled all the images containing annotated objects, resulting in 22 images in total. We then queried the GPT-4o [1] 231 times with each image pair annotated with object bounding boxes as visual prompts, the corresponding box coordinates, and the task prompt. By parsing the GPT-4o outputs, we obtained the results in Table 4.

In Table 4, the *model total* represents the performance of our model on the entire scene, and the *model adjusted* represents the performance evaluated only on those 22 subsampled images for a fair comparison. Besides the issues with computation cost and context limits, we note that a common failure pattern of VLM is the failure to maintain consistent object associations. The limitations of VLM in VPR are also discussed in the literature [43].

Table 4: **Pilot study for MLLM on MSG.** For the MLLM, we use GPT4o. The "model adjusted" is evaluated on the same set of images as the VLM.

| Metric | model total | model adjusted | VLM |
|--------|-------------|----------------|-----|
| PP IoU | 59.3 | 63.0 | 30.3 |
| PO IoU | 85.0 | 85.0 | 62.5 |

Nevertheless, this is only a small-scale pilot study. It is well possible to have better VLMs in the future and come up with better prompts, and we are excited about the future possibilities of MLLM + MSG.

## C.4 A qualitative real-world experiment

To examine how our method generalizes to real-world environments, we conducted a qualitative experiment. Specifically, We have self-recorded an unposed video with an iPhone in a household scenario and run our trained AoMSG model with a pretrained Grounding DINO detector on it.

In Figure 9, we show some resulting images with object instance ID labeled and a visualization of the generated graph. Results suggest that our model is able to obtain sensible outputs on arbitrary videos outside of the dataset.

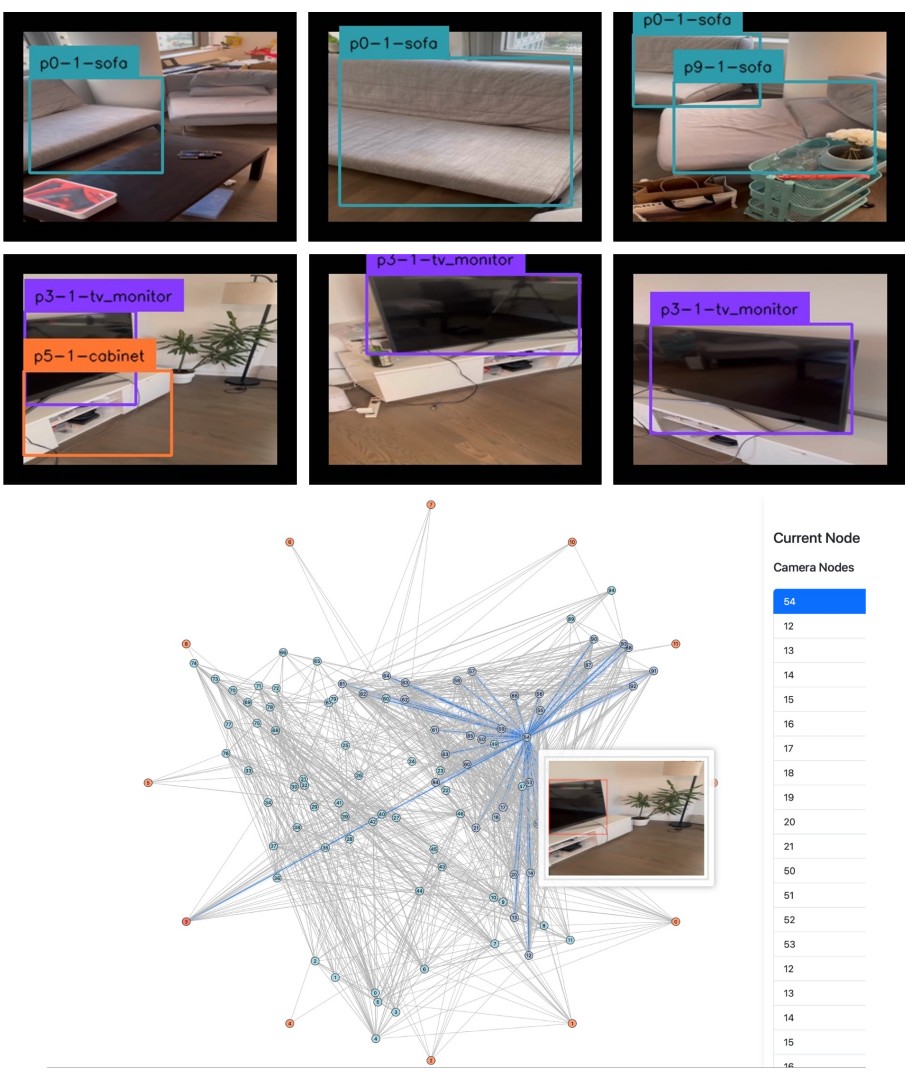

Figure 9: **Qualitative real-world experiment.** Top: results visualization. "px" is the object instance label. Bottom: a screenshot of the interactive graph visualization. Place nodes are in blue and object nodes are in orange.

## D   More visualizations

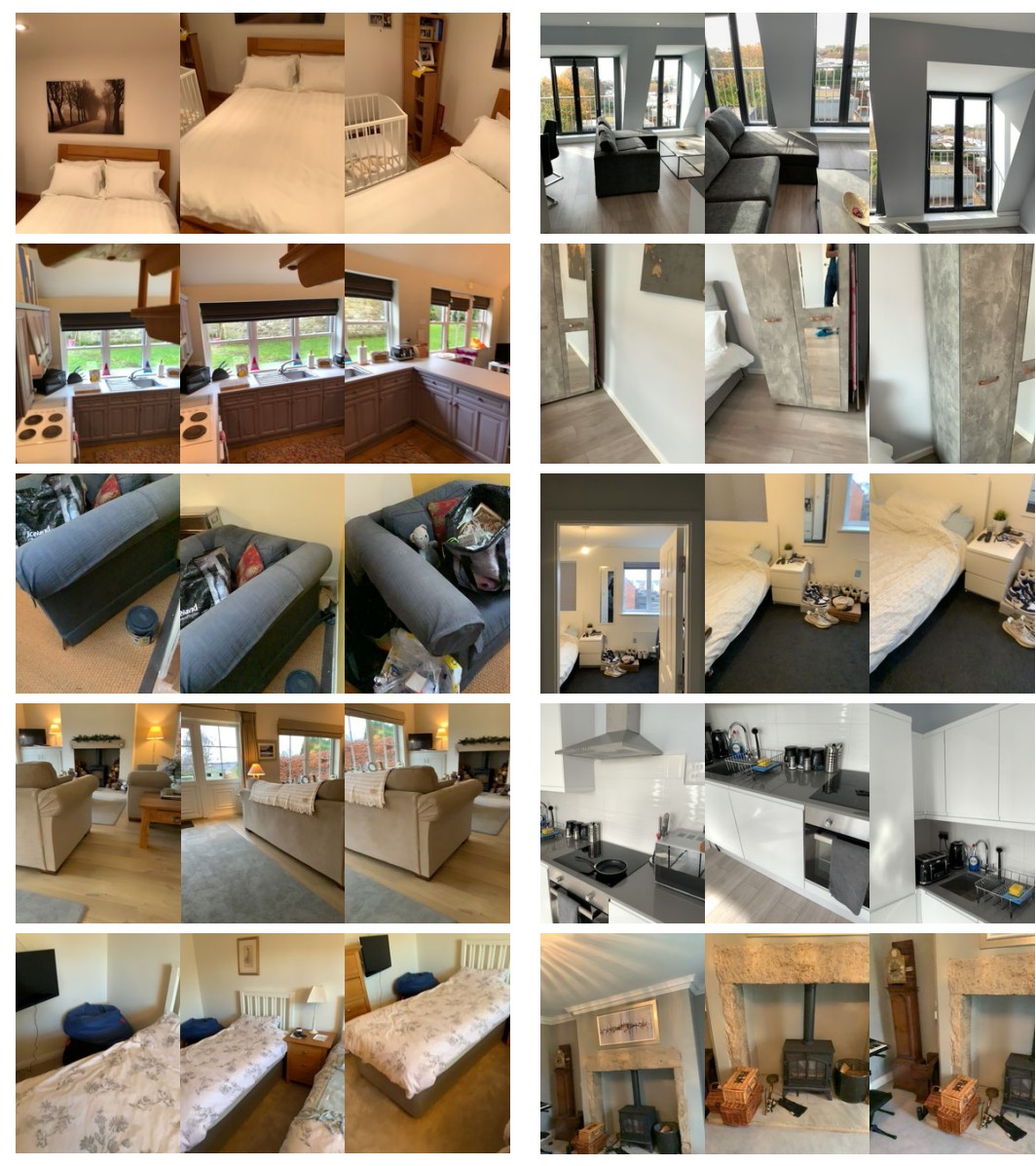

Figure 10: Visualization for the place nodes. Every 3 images shown side by side are those connected in the MSG, meaning they are considered from the same place.

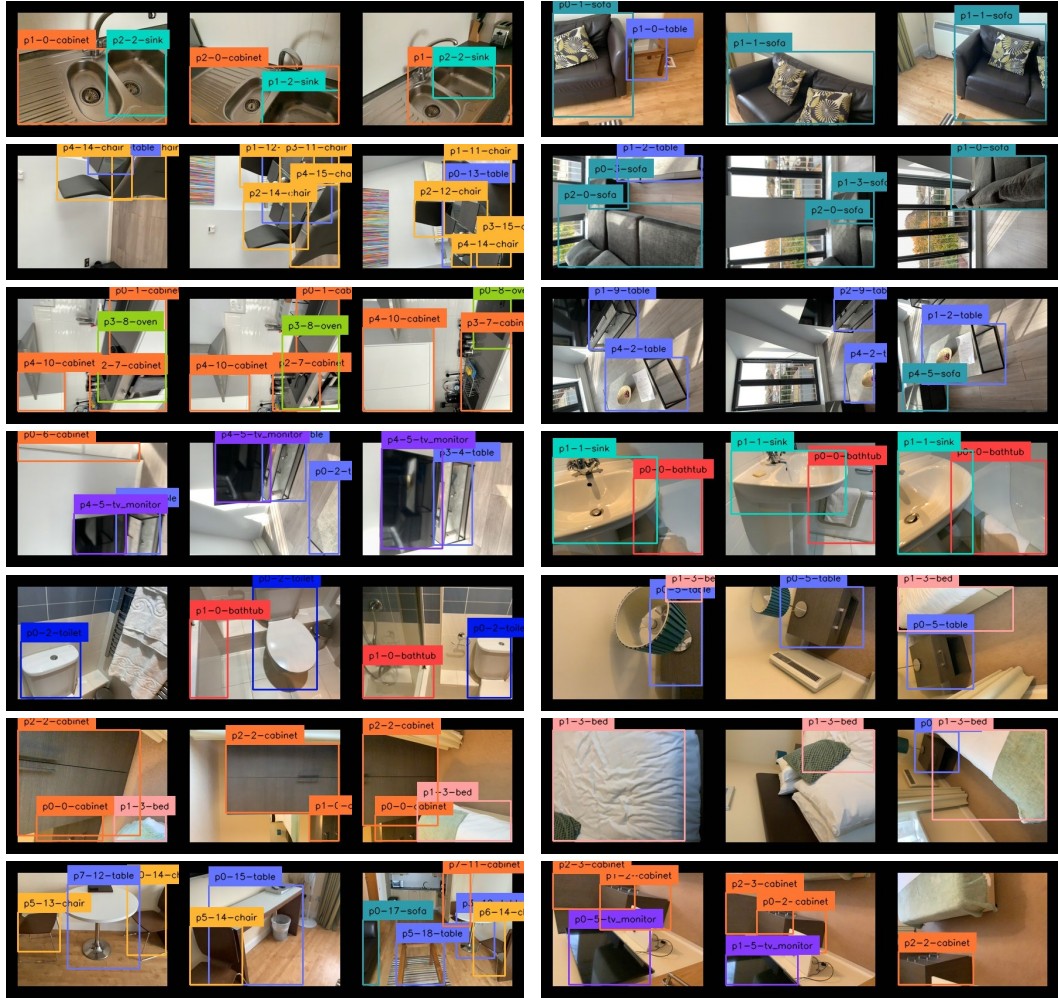

Figure 11: Visualization for the object nodes. The same objects recognized across different views are grouped as one object node. Each object is visualized with a colored bounding box with the annotation format: `predicted ID - groundtruth ID - groundtruth category`. For this visualization, we use the groundtruth detection bounding box in each frame as explained in the main paper. Note that images in some scenes are taken sideways, in the visualization we choose to keep it as is.

