# OpenReview forum: "Multiview Scene Graph"
_NeurIPS.cc/2024/Conference — NeurIPS 2024 poster_

### Official Review · Reviewer_o1XP · 2024-06-29

**Soundness:** 2
**Presentation:** 2
**Contribution:** 2
**Rating:** 5
**Confidence:** 3

**Summary:**

The paper introduces a new scene graph format, which regards objects and places as nodes. The edges are intuitive, basically saying which objects are in which places and which places are close to each other. The paper also provides two metrics on which the graph generation method can perform two tasks better than the chosen baselines. The authors also provide an application using the proposed graph to accelerate DUST3R's process.

**Strengths:**

The paper introduces a new scene graph format, which regards objects and places as nodes. The paper also provides two metrics on which the graph generation method can perform two tasks better than the chosen baselines.

**Weaknesses:**

### Motivation
Even though I understand what the authors are trying to express in the paper, the necessity of the proposed format of scene graph still raises my concern.
1. Firstly, the current scene graph extracted from multiple views is a bit simple; there is no spatial or other high-level semantic relationship between objects. From my point of view, the graph only associates the adjacent views for scene nodes and the same objects in the scene as object nodes. Do authors think one can leverage a VLM to do a similar job? Just query the VLM to associate the objects that appear in the multiple frames and query to associate the same scene if adjacent frames are close.
2. Secondly, do the authors think the proposed graph can somehow have a better ability to evaluate these tasks (VPR and object association)? If so, could you explain motivation more? What are the usages of edges between objects and places? Can existing scene graph formats, e.g., [1,2], do similar tasks in this paper?

### Experimental Issue
The experiments were not sufficient enough. The baselines in the paper were from early ages, and I suggest more recent ones should be compared. For example, two papers are first online in Nov 2023 [3] and in Feb 2024 [4]. Authors are encouraged to choose other methods as long as they are sensible to be recent baselines.

### Misc
1. Do the authors want to modify the title a bit? the current one seems to be a bit broad.
2. I may miss the related part, but what object detector did the authors use in Figure 2?
3. In the Supplementary, it is good to see the authors provide the source code, but the README is a void file, which makes it hard to understand the details.
4. The paper has several typos, e.g., "objects nodes" in Line 32 and "Simoutenous" in Line 72. Please do a careful review.

[1] Armeni I, He Z Y, Gwak J Y, et al. 3d scene graph: A structure for unified semantics, 3d space, and camera. CVPR 2019.

[2] Wu S C, Tateno K, Navab N, et al. Incremental 3d semantic scene graph prediction from rgb sequences. CVPR 2023.

[3] Izquierdo S, Civera J. Optimal transport aggregation for visual place recognition. CVPR 2024.

[4] Lu F, Lan X, Zhang L, et al. CricaVPR: Cross-image Correlation-aware Representation Learning for Visual Place Recognition. CVPR 2024.

**Questions:**

The same as above.

**Limitations:**

Limitations have been thoroughly discussed.

---

> ### Author Rebuttal · Authors · 2024-08-07
>
> __Q1__: the current scene graph extracted from multiple views is a bit simple
>
> __A1__: We acknowledge the fact that the proposed MSG has a rather simple format as it is free of spatial or high-level semantic relationships between objects.
> However, this does not mean this task is simple, insignificant, or can be readily solved. As acknowledged by reviewer 2 (AiA2) and demonstrated by the baseline performances in the paper, building topological connectivity between the unposed images and associating objects across frames and long stretches of time without observation is a challenging and important task. Constructing such multiview scene graphs from unposed images is fundamental to many tasks, such as 3D reconstruction[1], visual navigation[2], visual slam[3], etc. Having such a graph can be beneficial to all these applications.
>
> Furthermore, the proposed MSG is complementary to the existing scene graphs. Edges for object-object relationships can be a seamless add-on to extend the MSG with more semantic information. Therefore, we believe our work adds a meaningful contribution to the scene graph community.
>
> We also demonstrate in the next response that current Vision-Language Models (VLMs) are not yet capable of solving the MSG task.
>
>
> [1]Xiao, Jianxiong, Andrew Owens, and Antonio Torralba. "Sun3d: A database of big spaces reconstructed using sfm and object labels." Proceedings of the IEEE international conference on computer vision. 2013.
>
> [2] Chaplot, Devendra Singh, et al. "Neural topological slam for visual navigation." Proceedings of the IEEE/CVF conference on computer vision and pattern recognition. 2020.
>
> [3] Salas-Moreno, Renato F., et al. "Slam++: Simultaneous localisation and mapping at the level of objects." Proceedings of the IEEE conference on computer vision and pattern recognition. 2013.
>
>
>
>
> __Q2__: Do authors think one can leverage a VLM to do a similar job?
>
> __A2__: We appreciate the suggestion. VLMs exhibit strong emergent abilities for many tasks, and we agree it would be interesting to try them out for MSG. However, querying VLMs with every image pair is a huge amount of work and cost, and sending all images together poses a challenge to the context length limit while also hurting performance. Therefore, we conducted a case study with one scene as a pilot study.
>
> Specifically, we sampled a scene with a relatively small number of images and further subsampled all the images containing annotated objects, resulting in 22 images in total. We then queried the GPT-4o[1] 231 times with each image pair annotated with object bounding boxes as visual prompts, the corresponding box coordinates, and the task prompt. By parsing the GPT-4o outputs, we obtained the following results:
>
> | metric  | model total | model adjusted | VLM   |
> |---------|-------------|----------------|-------|
> | PP IoU  | 59.3        | 63.0           | 30.3  |
> | PO IoU  | 85.0        | 85.0           | 62.5  |
>
> The *model total* represents the performance of our model on the entire scene, and *model adjusted* represents the performance evaluated only on those 22 subsampled images for a fair comparison. Besides the issues with computation cost and context limits, we note that a common failure pattern of VLM is the failure to maintain consistent object associations. The limitations of VLM in VPR are also discussed in the literature [2].
>
> Nevertheless, we acknowledge that this is only a small-scale pilot study. It is well possible to have better VLMs in the future and come up with better prompts, and we are excited about the future possibilities of VLM + MSG.
>
> [1] Achiam, Josh, et al. "Gpt-4 technical report." arXiv preprint arXiv:2303.08774 (2023).
>
> [2] Lyu, Zonglin, et al. "Tell Me Where You Are: Multimodal LLMs Meet Place Recognition." arXiv preprint arXiv:2406.17520 (2024).
>
>
>
> __Q3__: do the authors think the proposed graph can somehow have a better ability to evaluate these tasks (VPR and object association)? … Motivation and usage. Can existing scene graph formats do similar tasks in this paper?
>
> __A3__: Thank you for the question. The proposed MSG encompasses both visual place recognition (VPR) and object association. Our proposed evaluation metrics, PP IoU and PO IoU, reflect the quality of place recognition and object association. Our proposed baseline combines the two tasks in a joint architecture. Therefore, we believe training and evaluating on the task of MSG will help both VPR and object association.
>
> We are motivated by the fact that building spatial correspondence by associating images and objects is fundamental for scene understanding. The MSG takes unposed RGB image sets as input, allowing it to be applied to any arbitrary image sets and video streams. As discussed in Q1, obtaining MSG from the unposed RGB images will provide a foundation for many downstream tasks in computer vision and robotics, such as loop closure, SfM, visual SLAM, navigation, and mobile manipulation.
>
> Existing scene graphs focus on describing the spatial and semantic relationships between different objects, which is different from the MSG task. Meanwhile, existing 3D scene graphs require having 3D scene representations before obtaining the scene graphs. This, where object association has been given, is already a good starting point for many vision tasks. However, such 3D scenes are not always available or can be reliably obtained from unposed RGB observations. Therefore, we believe our MSG is a complementary contribution to the existing scene graph literature.
>
> **Continue in the official comment**

---

> ### Author Response · Authors · 2024-08-07
> **Continue to answer reviewer's questions**
>
> __Q4__: Experiment. The baselines in the paper were from early ages
>
> __A4__: We thank the reviewer for this question. All the baselines are from recent years. The proposed MSG task involves two parts of baselines: visual place recognition (VPR) and object association. For the VPR part, our NetVlad baseline is adapted from the widely used baseline deep VG benchmark [1] (reference [8] in the manuscript), which was published in 2022. Anyloc was published in 2023. For the object association part, UniTrack is a widely used multi-object tracking method released in 2021. DEVA is from ICCV 2023.
>
> We appreciate the reviewer’s suggestion and have also evaluated optimal transport aggregation (Salad) [2] as an additional and more recent baseline for the VPR part. The results are reported below.  We find its performance comparable to Anyloc on the recall while slightly better on PP IoU. We note that both of the baselines are evaluated off-the-shelf.
>
> | model | Recall@1 | PP IoU |
> |---------------------------|-------|-------|
> | Anyloc                 | 97.1 | 34.2 |
> | Salad[2]               | 97.1 |  35.6 |
> | AoMSG-4	    | 98.3 | 42.2 |
>
> We would like to emphasize that the baseline methods we proposed (AoMSGs) are designed to be straightforward and easily extensible. Future work can incorporate more advanced techniques and insights from the VPR and object association fields, including but not limited to the Sinkhorn algorithm[3] used in the Salad paper. It would be exciting to explore more ways of combining the wisdom in both place and object association fields for MSG in future work.
>
> [1] Berton, Gabriele, et al. "Deep visual geo-localization benchmark." CVPR. 2022.
>
> [2] Izquierdo S, Civera J. Optimal transport aggregation for visual place recognition. CVPR 2024.
>
> [3] Marco Cuturi. Sinkhorn distances: Lightspeed computation of optimal transport. Advances in neural information processing systems, 26, 2013.
>
>
> __Q5__: Do the authors want to modify the title a bit?
>
> __A5__: Thank you for the question. We chose “Multiview Scene Graph” as the title because we believe it best describes the contribution of this work while maintaining conciseness. A multiview scene graph refers to our proposed task, where a place-object graph is inferred by associating unposed images across views, predicting their topological connectivity, and associating objects across multiple views simultaneously. So multiview association is the key characteristic and essence of the proposed scene graph and the task and we believe that naming the work “Multiview Scene Graph” highlights its core focus and makes it both memorable and easy to understand.
>
> However, we are more than happy to consider any advice on better titles from the reviewer.
>
>
> __Q6__: What Object Detector is used in figure 2?
>
> __A6__: Any detector can be used as long as it proposes object bounding boxes. The detector is frozen and bounding boxes are the only thing our method needs from the detector. For training, we use groundtruth detection for efficient supervision. We also used Grounding DINO as our off-the-shelf object detector for open-set detection and it shows consistent performance order for all the methods.
>
>
> __Q7__: The README is a void file in the supplementary materials.
>
> __A7__: Thanks for pointing it out. We deleted README while trying to keep anonymity. We will upload the complete code base and model checkpoint when our paper is camera-ready.
>
>
> __Q8__: typos.
>
> __A8__: We thank the reviewer for pointing this out. We will fix all typos for the camera-ready version.

---

> ### Author Response · Authors · 2024-08-12
> **Inquiry About Any Additional Concerns**
>
> Dear reviewer o1XP
>
> Thanks for your reviews, which are valuable for improving the overall quality of this manuscript.
>
> To address your concerns, we have added discussions and explanations of the definition and format of our proposed Multivew Scene Graph (MSG) and how it is different from other existing formats of scene graphs. We have also discussed the potential applications and benefits of the MSG.
>
> For the experiments, we appreciate your constructive suggestions and have conducted an additional baseline based on the recent VPR literature you recommended. We will also add these references to the revised version. We would also like to thank you for suggesting the idea of using VLM. We like this idea and conducted a pilot study. We have presented the results and discussion in the response above as well as in the attached PDF file.
>
> We have also provided answers and clarifications to the other questions of your concerns. We thank you for pointing out the typos and we promise to fix them in the revised version. We have also explained our motivation for choosing the title and we would love to hear from you for any advice on this.
>
> Could we kindly ask if our responses have addressed your concerns and if you have any new questions? Thanks for your time and effort in reviewing our submission.
>
> Authors of Submission ID 18923

---

> > ### Comment · Reviewer_o1XP · 2024-08-13
> > **Thank for the rebuttal**
> >
> > Hi, I am happy to see your rebuttal. I hope you can include adjustment in your final paper. I would like to change the rate to BA.

---

> > > ### Author Response · Authors · 2024-08-13
> > > **Thank you**
> > >
> > > Thank you for your valuable reviews and we are very happy to see your positive decision. Yes, we appreciate your constructive advice and we will surely include these adjustments in the revised final paper.

---

### Official Review · Reviewer_9wC7 · 2024-07-09

**Soundness:** 3
**Presentation:** 4
**Contribution:** 2
**Rating:** 4
**Confidence:** 3

**Summary:**

The paper proposes a novel task: generating scene graphs from unposed RGB images as well as a new benchmark based on ARKitscenes. To achieve this, the paper proposes a novel approach that use off-shelf image encoder and detector that are frozen and only train a decoder that takes in the features learned from frozen image encoder and detector to generate scene graphs. The author ablates the pre-trained weights for backbones and show a promising results compared to baselines.

**Strengths:**

1. The author proposes a new benchmark based on ARKitscenes, and a novel method trained on those, showing promising results.
2. the authors also ablate the influence of different pre-trained backbones.

**Weaknesses:**

1. The task is not that novel, see EgoSG: Learning 3D Scene Graphs from Egocentric RGB-D Sequences, where a very similar task is propose, as in, generating scene graphs from unposed RGB-D images. And from RGB to RGB-D, a off-shelf depth detector can be leveraged.

2. it is unclear, how was ViT pre-trained, it is using MAE?

3. the authors use a self-made baseline, i am wondering if other baselines could be compared to, such as RGB + depth predictor to simulate a RGB-D methods, or using the visual clues for pose estimations and using existing methods as baselines.

4. Lack of explanation of design of choices: why explicit leveraging detection predictions by cropping features? What if you just feed in the image features? How much does "explicit leveraging detection results" help?

5. Seeing the gap between w/ GT detection w/ GDino, it seems better detector is more important than the proposed detector itself, have you tried  more powerful detectors?

**Questions:**

1. do you need the unposed images must have overlap with each other?
2. Have you tried SAM? and do you think DinoV2 performs best is because of the detector's feature is closed to image encoder, when using DINOv2 as image encoder?

**Limitations:**

it is not that hard to get poses from either sensors or visual cues nowadays.

---

> ### Author Rebuttal · Authors · 2024-08-07
>
> __Q1__:The task is not novel, see EgoSG.
>
> __A1__: Thank you for the question. We will cite EgoSG as a related work. However, while both carry “scene graph” in the names, the MSG task is completely different from the EgoSG and other existing 3D scene graphs.
>
> Firstly, the definition of the graph is different. The existing scene graph connects objects and describes their relationships with the graph edges. We propose MSG as place-object graphs, where edges between images reflect the topological connectivity, and edges between objects and images reflect the object locations after resolving the object association. As reviewer 2 (AiA2)
>  acknowledges, associating objects across frames and long periods without observation is a challenging and important task that does not truly require 3D information.  We believe introducing and studying the multiview unposed scene graph will provide a meaningful topological scene representation beneficial to other downstream tasks in computer vision and robotics, such as loop closure, SfM[1], visual SLAM[2], visual navigation[3], and mobile manipulation.
>
> Secondly, the input data format is different. EgoSG and other 3D scene graphs require having 3D scene representations before obtaining the scene graphs, which is already a very good starting point for many vision tasks and is not always available. Our proposed MSG focuses on a different challenge. The input is purely unposed RGB images without any additional spatial knowledge. MSG exactly aims to extract topological spatial knowledge from these inputs. This allows it to work on arbitrary videos when depth information is not given or needs to be estimated and lays a good foundation for reconstructing 3D information from 2D inputs. While estimating depth and poses to convert the problem into RGB-D could improve performance, it does not solely solve our task as the estimation itself can introduce errors.  As suggested by the reviewer(discussed in the following response Q3A3), we have tried a recent strong pose estimation method [4] and found the performance less satisfactory due to the accumulated drifting pose errors. The loop closure provided by the MSG will in fact help better estimate poses.
>
> Additionally, our MSG will be complementary to the existing scene graphs. Edges for object-object relationships can be a seamless add-on to extend the MSG with more semantic information. Therefore, we believe our work adds a meaningful contribution to the scene graph community.
>
> In conclusion, we believe this is a novel task different from the previous scene graphs and also important for many downstream applications in computer vision and robotics. This paper contributes by introducing the MSG task, the set of new baselines, and benchmarks to stimulate future research.
>
> [1]Xiao, Jianxiong, Andrew Owens, and Antonio Torralba. "Sun3d: A database of big spaces reconstructed using sfm and object labels." ICCV. 2013.
>
> [2] Salas-Moreno, Renato F., et al. "Slam++: Simultaneous localisation and mapping at the level of objects." CVPR. 2013.
>
> [3] Chaplot, Devendra Singh, et al. "Neural topological slam for visual navigation." CVPR. 2020.
>
> [4] Barroso-Laguna, Axel, et al. "Matching 2D Images in 3D: Metric Relative Pose from Metric Correspondences." CVPR. 2024.
>
> __Q2__: how was ViT pre-trained:
>
> __A2__: Thank you for the question. We will update the manuscript with this information. For the ViT model, we are using the base model with the default weights pretrained on ImageNet-21K. It is not a masked auto-encoder (MAE).  MAE is also an important image pretraining model. We would like to note that any visual encoders that can produce feature maps or tokens are adaptable to our method. However, we did not sweep through more available choices as we found that DINOv2 gives significantly better performance.
>
> __Q3__: other baselines with existing methods.
>
> __A3__: Thank you for the suggestion. We agree adding more baselines linked to existing tasks and methods will benefit our work. Therefore, We have added a new baseline based on pose estimation. We are happy to explore other possible directions in the future. Specifically, We use a pretrained Mickey model [4] and provide the image data sequentially in temporal order and also the corresponding intrinsics ( not provided to MSG). We compute relative poses and convert them to absolute poses w.r.t the first frame. Then use the same threshold as the data set hyperparameter to obtain the p-p adjacency matrix and P-P IoU. The results are listed in the table below. We see that the performance is close to that of Anyloc. Recall@1 is easily 1.0 since the data is given in temporal order, and consecutive frames are trivially recalled. But its P-P IoU is inferior to Anyloc, suggesting the drifting issue of estimated poses and the need for loop closure, which is precisely MSG’s strength.
>
> | model | Recall@1 | PP IoU |
> |---------------------------|-------|-------|
> | Anyloc                 | 97.1 | 34.2 |
> | Mickey[1]              | 100 | 33.1 |
>
> [4] Barroso-Laguna, Axel, et al. "Matching 2D Images in 3D: Metric Relative Pose from Metric Correspondences." CVPR. 2024.
>
>
> __Q4__: why explicit leveraging detection predictions by cropping features? What if you just feed in the image features? How much does "explicit leveraging detection results" help?
>
> __A4__: The task of MSG involves associating objects from images across views. Therefore, we leverage detection bounding boxes to crop features to obtain features of object appearances. If directly feeding the image features as a whole, it would be difficult and unnatural for object association since the model would have no prior knowledge of what are the interested objects. So explicitly leveraging detection results will help the model locate the object appearances in the input images and associate them to build the multiview scene graph.
>
>
> **Continue in the official comment.**

---

> ### Author Response · Authors · 2024-08-07
> **Continue to answer reviewer's question**
>
> __Q5__: Seeing the gap between w/ GT detection w/ GDino, it seems a better detector is more important than the proposed detector itself, have you tried more powerful detectors?
>
> __A5__: Our MSG task and model tackle object association rather than object detection. Association happens after object detection and refers to identifying whether the object detected in one image is the same object detected in another. While it is true that a better detector will enhance the overall performance of MSG, it is inaccurate to say that the detector is more important than MSG and object association itself, as they focus on different tasks.
>
> The gap between ground truth detection and Grounding Dino (GDino) is expected since GDino is used off-the-shelf and in a zero-shot manner. Our aim is to demonstrate that although MSG uses ground truth detections during training, any available detector can be incorporated into the framework, and the performance order remains consistent. Thus, we anticipate that a more powerful detector than GDino or one fine-tuned on the same data should yield better results.
>
> The task focuses on associating places and objects, and we believe having ground truth detections and consistent results with the commonly used GDino is sufficient to convey our contributions. We appreciate the reviewer’s suggestion and will consider training or benchmarking detectors in future explorations.
>
>
> __Q6__: if overlapped unposed images are needed:
>
> __A6__: No, we do not make such overlapping assumptions for the unposed images. The input of the MSG task is just a set of unposed RGB images. The model learns to figure out whether the images are taken from the closeby positions and associate the object appearances, i.e. whether they are the same object. The input of the entire process is simply unposed RGB images, with no overlapping required.
>
>
> __Q7__: Have you tried SAM? do you think DinoV2 performs best because the detector's feature is close to the image encoder when using DINOv2 as the image encoder?
>
> __A7__: Yes, our baseline DEVA uses SAM to obtain object segmentations and perform video segmentation by tracking these object segments across the frames. Detector’s features are not used in our model. The detector only provides bounding boxes. Grounding Dino is different from DINOv2 although bearing a similar name. The former is a Transformer-based detection model and the latter is a Transformer-based self-supervised vision pretraining model. DINOv2’s features have shown great performance in downstream tasks, such as in Anyloc. In this work, we have a similar observation that DINOv2 as the encoder backbone produces the best performance. We think the reason for its strong performance is possibly due to its vast and diverse pretraining data.

---

> ### Author Response · Authors · 2024-08-12
> **Inquiry About Any Additional Concerns**
>
> Dear reviewer 9wC7
>
> Thanks for your comments, which are valuable for improving the overall quality of this manuscript.
>
> To address your concerns, we have provided a thorough discussion and explanation of similarities and differences between our proposed Multivew Scene Graph (MSG) and other existing formats of scene graphs such as EgoSG. We really appreciate your thoughts and will include this in our references as well as the discussion of it in the related works section in the revised version. We have also conducted an additional baseline based on pose estimation for the place recognition part of MSG, thanks to your suggestion.
>
> We have also provided a thorough explanation to the other questions of your concerns, such as the use of the detectors in our proposed method, the design choices, encoder choices, and image choices.
>
> Could we kindly ask if our responses have addressed your concerns and if you have any new questions? Thanks for your time and effort.
>
> Authors of Submission ID 18923

---

### Official Review · Reviewer_AiA2 · 2024-07-10

**Soundness:** 3
**Presentation:** 3
**Contribution:** 3
**Rating:** 6
**Confidence:** 4

**Summary:**

The manuscript proposes the problem of inferring a scene graph from unposed images of a space. The key distinguishing factor to previous work is using multiple frames (as opposed to a single frame) and not requiring poses and depth (like for typical metric 3d scene graphs). The scenegraph is defined as a the set of images connected by edges implying topological connectedness and the set of objects. Objects are connected to place nodes based on where they are observed from. The manuscript also proposes a sensible baseline that outperforms more basic baselines for MVSG generation as well as methods that just work on place recognition or just on object tracking.

**Strengths:**

Associating objects across frames and long stretches of time without observation is a challenging and important task that does not truly require 3D information (but is greatly helped by it). Introducing and studying the relaxed problem of multi-view unposed scenegraph generation has the potential to yield novel solutions that can be deployed on any video stream. And of course methods that utilize 3d information (say from running some deep slam on the video first) might yield the best performance. It puts the different approaches onto the same benchmark which is useful for the community.

Given the 3D training data (to supervise associations correctly) the proposed model is conceptually and implementation wise simple and leverages state of the art vision backbones. Based on the evaluation this simple model is also effective in generating scene graphs from unposed images. The SepMSG baselines are useful in convincing about the usefulness of the AoMSG model. It is interesting to see that this baseline model outperforms existing tracking methods by a large margin.

Overall the paper is well written and the illustrations are high quality to support the understanding of the method and ideas.

**Weaknesses:**

The constraint of not requiring poses and theoretically enables the method to be run on any video sequence. However there are no examples of even qualitative experiments on some arbitrary video sequences. I think this should be easy to add and would help support the idea that unposed-image posed scene graphs are useful. The other aspect that is not addressed in the manuscript is how to train the proposed system without access to poses and 3d surfaces  or 3d bounding boxes of objects in order to supervise the model. It is unclear if the place localizer just learns to replicate the pose thresholds used during training (see question).

**Questions:**

- l 204 whats the "benchmark VG pipeline" ?
- SepMSG: I assume you are also using the same 2d encoder as for AoMSG? Could you clarify.
- The place localization is trained using relative pose thresholding to obtain positive samples. I am curious if that means the features just learn those spatial thresholds? During inference what is the relative pose distribution of the connected places? And the disconnected ones?

**Limitations:**

Limitations are addressed adequately in the paper.

---

> ### Author Rebuttal · Authors · 2024-08-07
>
> __Q1__: Example of qualitative experiments on some arbitrary video sequences.
>
> __A1__: Thank you for the great suggestion! We have self-recorded an unposed video with an iPhone in a household scenario and run our trained AoMSG model with a pretrained Grounding DINO detector on it. In the attached PDF file we show some resulting images with object instance id labeled. Results suggest that our model is able to obtain sensible outputs on arbitrary videos outside of the dataset. We have also made a simple frontend tool for interactive visualization with MSG, which we show a screenshot in the PDF. We will release the tool along with the full source code when camera-ready.
>
> __Q2__: How to train the model without poses or other 3D annotations.
>
> __A2__: Thank you for the question. Like many other new tasks, certain requirements of annotation exist in the current MSG task. We rely on explicit camera poses and 3D object annotations in the ARKitScene to generate our dataset since marking the same places and objects requires knowing camera poses and object instances. We note that any 3D dataset or environment providing such annotations can be leveraged for the MSG task and there are many large-scale 3D datasets available for use such as ScanNet [1], ScanNet++[2], and HM3D[3]. Nevertheless, we acknowledge that this is a limitation and in the future, we will try to explore training on a combined recipe of different datasets for strongly generalizable performance or explore training with less annotation.
>
> [1] Dai, Angela, et al. "Scannet: Richly-annotated 3d reconstructions of indoor scenes." Proceedings of the IEEE conference on computer vision and pattern recognition. 2017.
>
> [2] Yeshwanth, Chandan, et al. "Scannet++: A high-fidelity dataset of 3d indoor scenes." Proceedings of the IEEE/CVF International Conference on Computer Vision. 2023.
>
> [3] Ramakrishnan, Santhosh K., et al. "Habitat-matterport 3d dataset (hm3d): 1000 large-scale 3d environments for embodied ai." arXiv preprint arXiv:2109.08238 (2021).
>
>
> __Q3__:  What is the benchmark VG Pipeline
>
> __A3__: We thank the reviewer and apologize for the confusion. VG stands for Visual Geo-localization[1] which refers to the same task of Visual Place Recognition (VPR). This pipeline is referenced as [8] in the manuscript. We will revise this line in the manuscript for better clarity.
>
> [1] Berton, Gabriele, et al. "Deep visual geo-localization benchmark." Proceedings of the IEEE/CVF Conference on Computer Vision and Pattern Recognition. 2022.
>
> __Q4__: Encoder of SepMSG
>
> __A4__: Thank you for the question. To clarify, SepMSG baselines use the same image encoder as the AoMSG methods. We use pretrained encoders and keep them frozen while training all the methods. Specifically, SepMSG-direct refers to directly evaluating the output feature vector of the encoders, SepMSG-linear/MLP trains linear probing or MLP on top of the encoders and AoMSG trains the Transformer-based decoder. We empirically find that the pretrained DINO-base model produces the best performance in direct evaluation, so all the experiments in Table 1 use it as the baseline. Below is a table for the detailed performance of different backbone choices when used in SepMSG-direct. We will revise the manuscript with better clarification and the results.
>
> | Encoder for SepMSG-direct | PP IoU | PO IoU |
> |---------------------------|-------|-------|
> | ResNet 50                 | 26.30 | 45.23 |
> | ConvNext base                 | 27.62 | 46.29 |
> | ViT base                      | 28.86 | 46.04 |
> | DINO small                | 29.54 | 50.37 |
> | DINO base                 | 30.94 | 54.78 |
> | DINO large                | 31.02 | 50.02 |
>
>
> __Q5__: if features just learn those spatial thresholds? Calculate the relative pose distribution
>
> __A5__: Thank you for the great suggestion. Annotating based on spatial thresholds is a conventional setup in visual place recognition (VPR) tasks and we chose to follow this convention.  While training the features, we adopt a metric learning approach instead of simple binary prediction. During training, embeddings positive pairs are drawn closer in terms of cosine similarity, while negative pairs are pushed further. We also monitor the training dynamic using the Total Coding Rate as described in [1] to alert possible feature collapse. Thus, the learned features are not confined to just those spatial thresholds.
>
> We have also added 4 plots for relative pose distribution (both orientation and translation) in the attached PDF file. They are obtained by taking the model’s predictions on the test set and calculating the histograms of the relative orientation and translation differences of the predicted connected and not-connected places. From the plots, we can see that while the model makes some mistakes (currently the PP IoU is around 0.40), the distribution difference between the connected and the not-connected places is clear. There is a clear yet smooth separation at the spatial thresholds on all the plotted distributions.
>
> [1] Yu, Yaodong, et al. "Learning diverse and discriminative representations via the principle of maximal coding rate reduction." Advances in neural information processing systems 33 (2020): 9422-9434.

---

> > ### Comment · Reviewer_AiA2 · 2024-08-09
> >
> > Thank you for addressing my questions and comments. I appreciate the histograms showing the learned relative pose separation based on semantic features.
> >
> > The qualitative experiments on iPhone video is good to see although the limited set of frames makes it hard to judge how well the model was able to re-associate and track objects. It seems there are 12 objects - it would be good to see which ones were identified?
> >
> > I also had a look at the other reviews; I do agree that EgoGS using RGBD is a related work worth citing. I do think that the work of the authors has a potential for more impact since it can be run on any video sequence without additional processing (as demonstrated by the iPhone video experiment). While we could run a monodepth network, it is unclear how the typical problem of inferring consistent scale will impact the performance of EgoGS with monodepth.
> >
> > I was also excited to see the VLM comparison. It supports the need for dedicated approaches (at least thus far) that can run on long video sequences.
> >
> > All in all I am happy to stick with my rating.

---

> > > ### Author Response · Authors · 2024-08-12
> > >
> > > Dear reviewer AiA2,
> > >
> > > Thank you for your response. We are glad that you appreciate our work and the additional experiments. For the qualitative experiments, we have generated a whole video with object bounding box visualizations that can show the tracking and re-association effects. But we could only show a few frames in the attached PDF due to space limit. We definitely agree that more frames will be a better visualization and We are releasing more visualization videos on a website along with the interactive visualization tool after the anonymous phase.
> > >
> > > As for showing which objects are identified, that is an essential feature and we currently support it in the front-end visualization tool in the following ways:
> > >  - a) the orange nodes are object nodes, and hovering the mouse over it will show the object category and an object ID corresponding to the per-frame visualization, which has an object category and an object id labeled to the bounding box.
> > >  - b) when clicking, some edges will be highlighted in blue (as you can see in the attached screenshot in the PDF). Those between the orange nodes and the blue nodes indicate which and where an object is identified. We are excited to release it for more interactive visualization and we are super happy if you could provide more suggestions so that we can make it better.
> > >
> > > Please do not hesitate if you have more questions or suggestions. Thank you again for your time and efforts.

---

### Official Review · Reviewer_qE49 · 2024-07-13

**Soundness:** 3
**Presentation:** 3
**Contribution:** 2
**Rating:** 6
**Confidence:** 3

**Summary:**

The paper introduces the novel task of Multiview Scene Graph generation from unposed RGB images, whereas this type of scene graph encodes the notion of 'places', i.e., images from spatially close locations, and detected objects as graph nodes. The motivation is to combine spatial reasoning of object association and place recognition into a joint representation.

An evaluation metric based on the edge adjacency is introduced along an initial baseline method (AoMSG, based on a Transformer architecture and off-the-shelf image encoder and object detector) for this new task, which is compared against sub-task specific methods on the ARKit scenes dataset.

**Strengths:**

The paper presents and motivates a new task of building a topological scene graph combing place recognition and object association. An example application task (3D Reconstruction using Dust3r on sub-sets of input images retrieved from the MSG) which would benefit from the proposed MSG is described in the discussion section.
The proposed adjacency-IoU metric is simple, however the object-object "alignment" can play a dominant role.

**Weaknesses:**

- The current task and proposed baseline model does not seem to allow for an adjustable "place" definition beyond the train set constraints (place = 1m and 1 rad).
- To give further insights about the implication of the proposed metric, it would be useful to provide failure case which results in low IoU scores, especially for PP IoU which is significantly lower than PO IoU.

**Questions:**

- Missing experimental details of Figure 3: How do the numbers provided in the graph relate to the numbers in Table 1, i.e., which model specifically was trained with the specific backbone?

**Limitations:**

The checklist is complete, including justifications for each item. Limitations of the work are discussed in the paper.

---

> ### Author Rebuttal · Authors · 2024-08-07
>
> __Q1__: The current task and model do not seem to allow for an adjustable "place" definition beyond the train set constraints
>
> __A1__: Thank you for the question. The thresholds are dataset hyperparameters which is a conventional setup in visual place recognition (VPR) tasks and datasets [1, 2]. Since our MSG task involves place recognition, we choose to adopt this convention. VPR tasks require a model to classify whether two images are taken from the same place or not. The concept of “place” is a discretization of the space which is continuous by nature, which necessitates the use of thresholds in the VPR setup.
>
> The proposed baseline model conducts supervised training, so it is possible to set different hyperparameters for different datasets and train the model on multiple datasets. To give a closer look at the effect the threshold has on the model, in the attached PDF file, we also included figures of relative pose distributions (orientation and translation) for the connected and non-connect nodes based on our model’s prediction. The figures show that instead of collapsing to only represent the fixed thresholds, the pose distributions have a clear yet smooth separation across the spatial thresholds.
>
> [1] Lowry, Stephanie, et al. "Visual place recognition: A survey." ieee transactions on robotics 32.1 (2015): 1-19.
>
> [2] Zaffar, Mubariz, et al. "Vpr-bench: An open-source visual place recognition evaluation framework with quantifiable viewpoint and appearance change." International Journal of Computer Vision 129.7 (2021): 2136-2174.
>
> __Q2__: Give further insights about the implication of the proposed metric … provide failure cases that result in low IoU scores.
>
> __A2__: Thank you for the suggestion. We have added visualizations of some failure cases in the attached PDF file. We observe that most failure cases can be attributed to having very similar visual features with relatively large pose differences (false positives), such as observing a room from two opposite sides, or having fewer similar visual features with relatively smaller pose differences (false negatives).
> We note that the recall metric, conventional in VPR, is straightforward and effective for image retrieval against a database. However, it falls short in reflecting challenging false positives and negatives, especially when constructing a topological graph like MSG where the number of positives varies. This highlights the usefulness of our proposed IoU metric, which consistently evaluates the quality of the graph.
>
>
>
> __Q3__: Missing details in Figure 3.
>
> __A3__: Thank you for pointing it out. We made a mistake with the numbers and y-axis. We have uploaded a new figure with more information in the attached PDF file to replace the original one. The numbers marked as *direct* are obtained by directly evaluating the pretrained encoders corresponding to the SepMSG-direct baseline. Those marked as *AoMSG* are obtained by training the *AoMSG-4* model on top of the frozen encoders, following the same setup as the main experiments in Table 1 in the manuscript. We chose the DINOv2 base as the default encoder for our main experiments as it gives the best performance when evaluated directly. We will revise the manuscript with the updated Figure 3.

---

> > ### Comment · Reviewer_qE49 · 2024-08-12
> >
> > Dear Authors,
> >
> > thank you for providing the additional results regarding failure cases and their interpretation - this helps in understanding the metric and potential cause of errors in applying the method.
> >
> > Following the initial question of fellow Reviewer o1XP and the comment  AiA2, I was also excited about the provided VLM comparison and the Author's effort in running such an experiment given the limitation (e.g., context length) of current public VLM services.
> >
> > I'm happy to stick to my initial rating.

---

> > > ### Author Response · Authors · 2024-08-12
> > > **Thank you**
> > >
> > > Dear reviewer qE49,
> > >
> > > Thank you for your valuable reviews and we are pleased to see your positive decision. We really appreciate your constructive suggestions and we will include them in the revised version.

---

### Author Rebuttal · Authors · 2024-08-07

We thank our reviewers for their encouraging comments, helpful suggestions, and insightful questions.

**Acknowledgements**.
-  All reviewers acknowledge our novelty.
   -  R1 qE49: “The paper introduces the novel task…”.
   -  R3 9wC7: “The paper proposes a novel task…” “proposes a new benchmark … a novel method … showing promising result”.
   -  R4 o1XP: “The paper introduces a new scene graph format…”.
   -  R2 AiA2 further acknowledges that our posed MSG “is a challenging and important task” and “is useful for the community”.
- Several reviewers also think the paper is well-written with good presentation quality.
   -  R2 AiA2: “Overall the paper is well written and the illustrations are high quality to support the understanding of the method and ideas.”
   -  R3 9wC7 rates “Presentation: 4: excellent.”
We thank all the reviewers for their appreciation of this work.

**Questions and suggestions**.

Reviewers’ questions and suggestions are very constructive. We sincerely thank them for their advice in making the work of better quality.
   -  R1 qE49 and R2 AiA2 suggest adding additional analysis of the relative pose distributions and failure cases. We added these results in the attached PDF file and provided our thoughts and analysis in the responses.
   -  R3 9wC7 and R4 o1XP suggest additional baselines. We implement them and report the results and analysis in the PDF file as well as in the responses.
   -  R3 9wC7 and R4 o1XP also raise questions regarding the similarities and differences between our proposed Multivew Scene Graph (MSG) and other existing formats of scene graphs, to which we explain the difference in the definition and format of the graph, the input data, and the downstream applications. We also note that the MSG is complementary to the existing scene graphs. We answer the question in more detail in the corresponding responses.

**Attached PDF file**.

In total, we collected reviewers' constructive feedback and added the following 6 qualitative and quantitative experiments:
1. An analysis of relative pose distribution from the model’s prediction.
2. Provides failure cases which results in low IoU scores.
3. A qualitative real-world experiment on self-recorded videos using an iPhone.
4. Additional baseline using the most recent pose estimation method.
5. A pilot study of using VLM to try to solve the proposed MSG task.
6. Additional baseline using the most recent VPR method.

Please find the details in the PDF file and the responses. We will also revise the manuscript to reflect these experiments when the paper is camera-ready.

**Typos and revisions**.

We thank all the reviewers for pointing out the typos and confusion. Per the rules of the rebuttal session, we have marked all the typos and revisions suggested by the reviewers and will revise the manuscript when camera-ready.

---

### Decision · Program_Chairs · 2024-09-25

**Decision:**

Accept (poster)

**Comment:**

This paper proposes a methodology to build Multiview Scene Graphs from a set of unposed images. The paper initially received discordant ratings (2x BR, 2x WA). After rebuttal and discussion, the two positive reviewers confirmed their initial ratings (WA) and one of the two BR decided to raise their rating to BA.
The 4th reviewer was unresponsive and did not participate to the discussion. Given their initial rating is borderline and we don't know how the rebuttal has met their initial concerns, their initial rating is not taken into account for the final decision.
The final AC recommendation is thus to accept this work as poster.